# Utilizing Deep Feature Fusion for Automatic Leukemia Classification: An Internet of Medical Things-Enabled Deep Learning Framework

**DOI:** 10.3390/s24134420

**Published:** 2024-07-08

**Authors:** Md Manowarul Islam, Habibur Rahman Rifat, Md. Shamim Bin Shahid, Arnisha Akhter, Md Ashraf Uddin

**Affiliations:** 1Department of Computer Science and Engineering, Jagannath University, Dhaka 1100, Bangladesh; manowar@cse.jnu.ac.bd (M.M.I.); b170305028@cse.jnu.ac.bd (H.R.R.); b170305022@cse.jnu.ac.bd (M.S.B.S.); arnisha@cse.jnu.ac.bd (A.A.); 2School of Info Technology, Deakin University, Burwood, VIC 3125, Australia

**Keywords:** leukemia, VGG16, DenseNet-121, segmentation, feature fusion, transfer learning, internet of medical things

## Abstract

Acute lymphoblastic leukemia, commonly referred to as ALL, is a type of cancer that can affect both the blood and the bone marrow. The process of diagnosis is a difficult one since it often calls for specialist testing, such as blood tests, bone marrow aspiration, and biopsy, all of which are highly time-consuming and expensive. It is essential to obtain an early diagnosis of ALL in order to start therapy in a timely and suitable manner. In recent medical diagnostics, substantial progress has been achieved through the integration of artificial intelligence (AI) and Internet of Things (IoT) devices. Our proposal introduces a new AI-based Internet of Medical Things (IoMT) framework designed to automatically identify leukemia from peripheral blood smear (PBS) images. In this study, we present a novel deep learning-based fusion model to detect ALL types of leukemia. The system seamlessly delivers the diagnostic reports to the centralized database, inclusive of patient-specific devices. After collecting blood samples from the hospital, the PBS images are transmitted to the cloud server through a WiFi-enabled microscopic device. In the cloud server, a new fusion model that is capable of classifying ALL from PBS images is configured. The fusion model is trained using a dataset including 6512 original and segmented images from 89 individuals. Two input channels are used for the purpose of feature extraction in the fusion model. These channels include both the original and the segmented images. VGG16 is responsible for extracting features from the original images, whereas DenseNet-121 is responsible for extracting features from the segmented images. The two output features are merged together, and dense layers are used for the categorization of leukemia. The fusion model that has been suggested obtains an accuracy of 99.89%, a precision of 99.80%, and a recall of 99.72%, which places it in an excellent position for the categorization of leukemia. The proposed model outperformed several state-of-the-art Convolutional Neural Network (CNN) models in terms of performance. Consequently, this proposed model has the potential to save lives and effort. For a more comprehensive simulation of the entire methodology, a web application (Beta Version) has been developed in this study. This application is designed to determine the presence or absence of leukemia in individuals. The findings of this study hold significant potential for application in biomedical research, particularly in enhancing the accuracy of computer-aided leukemia detection.

## 1. Introduction

Leukemia is a malignancy characterized by the uncontrolled growth of abnormal white blood cells, leading to a life-threatening condition. It is a prevalent and devastating disease that can afflict individuals across all age groups, including both pediatric and adult populations and is a major contributor to global mortality rates. Leukemia affected 2.3 million individuals globally in 2015, and it was responsible for 353,500 deaths. Acute lymphoblastic leukemia (ALL), acute myeloid leukemia (AML), chronic lymphocytic leukemia (CLL), and chronic myeloid leukemia (CML) are the four most prevalent kinds of leukemia, and in both adults and children, acute lymphocytic leukemia (ALL) is a deadly tumor; it is responsible for approximately twenty-five percent of all child cancers [1]. According to the American Cancer Society [2], roughly 59,610 people were diagnosed with leukemia this year and 23,710 lost their lives in the USA among them. A total of 20,380 new cases of acute myeloid leukemia (AML) and 11,310 fatalities were reported in 2023. ALL is a rapidly progressing blood cancer that targets and can affect the bone marrow, blood, and other important organs including the liver, brain, kidney, etc. quickly, and if not discovered and treated promptly, it can lead to death within months [3,4].

If early diagnosis of ALL is achieved, it will lead to the prompt beginning of treatment, resulting in a substantial improvement in patient survival rates. A range of treatment options, such as chemotherapy, radiotherapy, medications for cancer, or a combination, might be considered based on the symptoms of the patient and threat level. Traditionally, hematologists examine blood smears or bone marrow samples for identification, and this diagnostic procedure involves the use of a microscope and relies heavily on the expertise and experience of pathologists. The process of manually detecting hematological disorders is laborious, costly, and time-intensive.

The potential of Internet of Things (IoT) to optimize medical procedures and improve patient care is what renders it so significant in the healthcare industry. By means of data-driven decision-making, efficient resource management, remote patient surveillance, and IoT, frequent hospital visits are eliminated, resulting in cost savings [5]. By connecting medical devices, implants, and wearables, IoMT establishes a network for the continuous collection of health data. AI-powered real-time analysis expedites disease diagnosis via remote monitoring and assignment simplification. The incorporation of Internet of Things (IoT) into the healthcare sector enables predictive analytics, improves diagnostic precision, facilitates telehealth services, and contributes to enhanced accessibility and proactive healthcare measures. IoT optimizes resource utilization, enhances patient outcomes, and lessens the financial burden associated with healthcare delivery. Security, interoperability, and accessibility, on the other hand, must be considered in order to ensure a responsible and equitable implementation [6]. Recently, clinical diagnosis and evaluation have been making significant progress as a result of advancements in clinical and microscopic image processing, leading to the development of algorithms and methodologies [7]. Notably, AI methods, such as machine learning and deep learning, enable the development of more precise algorithms for diagnosing blood disorders using appropriate image analysis [8,9]. For example, the authors utilized the CLAHE algorithm to improve image contrast and quality, and the leukocytes were extracted using color-based k-means clustering. For texture extraction, the approach uses the Gray-Level Co-occurrence Matrix (GLCM) and GLRLM. This work performed well with an accuracy of 96% using the SVM method with a Radial Basis Function (RBF) kernel [10]. The authors’ proposed approach that utilizes a CNN based transfer learning model VGG16 and Efficient Channel Attention (ECA) aims to enhance the extraction of high-quality deep features from the image dataset, which leads to improved feature representation and more accurate classification outcomes with an accuracy rate of 91.1% [11]. A pre-trained customized AlexNet was developed to classify ALL while overfitting was reduced via data augmentation, and this research attained a level of accuracy up to 96.06% [12]. Segmentation is a fundamental preprocessing step, and it is essential for enhancing specific features of interest. By isolating and focusing on relevant regions, we can apply feature enhancement techniques more effectively, leading to improved data analysis, visualization, and decision-making in a wide range of applications. Various segmentation methods have been applied to detect ALL [13,14]. In their study, Acharya et al. [15] employed k-medoid segmentation that demonstrates superior performance in segmentation and demonstrated an accuracy percentage of 98.60%.

Image fusion using segmentation and the original image in deep learning is a technique that combines information from two or more images and retains important details and features from each source. This can be particularly useful in medical imaging and object recognition [16,17,18,19]. In this research, an AI-based IoMT framework is proposed, where we concatenate features of original and segmented images to train and classify acute leukemia cancer. Two well-known deep learning models are deployed and customized for feature extraction. The DenseNet-121 [20] model is employed to extract the features from the segmented images while VGG16 [21] is customized with the original ALL images. The features of both models are then combined for a better understanding of the feature map, and these combined features are then fed into a CNN network to train and classify the images. The proposed model can achieve a higher accuracy that outperforms the related literature described in the related literature section. In order to direct our research, we developed a set of research questions:RQ-1: What are the benefits of IoMT devices and cloud servers in ALL detection?RQ-2: What are the impacts of combining the features from the original image and the segmented images?RQ-3: Does the feature fusion improve the performance results of the proposed model?RQ-4: Can the approach surpass the current literature in terms of accuracy for detecting leukemia cancer?

The initial research inquiry pertains to IoMT devices and cloud servers, which have a significant impact on the medical sector due to their ability to simultaneously reduce the time and cost of medical tests. In addition, medical toolkits are more expensive than IoMT devices and cloud servers. Using the cloud server, we can run the deep learning model which can provide the test results to the medical center and also to the patient’s devices. To address the other research questions, we utilized the features from original and segmented images and combined them for better insights and understanding of the image dataset. We ensured that no features were lost throughout the procedure. Before feeding the features to a condensed CNN for classification, we trimmed the DenseNet-121 and VGG16 models with consistent image dimensions during extraction to maintain feature uniformity. The evaluation of the proposed model was demonstrated for both original and segmented images to visualize the versatility of the model. Several important metrics were used to evaluate the performance of the current research. We discovered that the present study, where we utilized original and segmented image features combined, can detect leukemia more accurately compared to the existing works, making it applicable to automated computer-assisted leukemia cancer diagnosis in healthcare centers that can effectively save diagnosis time and costs.

In summary, the present research makes the following contributions:The key contribution of this proposal is a fusion model situated in a cloud server, which is capable of classifying acute lymphoblastic leukemia (ALL) from hematogone instances and determining ALL subtypes. The model achieves outstanding performance, demonstrating its efficacy in leukemia categorization. With an accuracy of 99.89%, the model excels in accurately identifying leukemia cases. The high performance indicates the reliability and effectiveness of the proposed framework.This research reveals that the approach of combining diverse image features, such as the original and segmented images, contributes to the overall success of the proposed fusion model in leukemia classification. When using the fusion model, the accuracy surpasses models trained solely on original or segmented images. The findings illustrate the importance of leveraging complementary information from different image sources to improve the model’s ability to detect patterns effectively.The proposed methodology operates within a structured framework based on Internet of Medical Things (IoMT). To provide a more comprehensive insight into this study, a demonstration of the proposed methodology is presented through a testbed implementation showcased in a web application. The AWS cloud service is used for image storage, segmented image generation, and prediction report transmission to patient devices in this demonstration.

We outline our studies as follows: The relevant research is outlined in Section 2. Section 3 details the data investigation, augmentation, and proposed frameworks. Section 4 presents the experimental results, while Section 5 and Section 6 conclude the study with a discussion and propose future research directions.

## 2. Literature Review

This section highlights the relevant existing studies on leukemia disease classification utilizing several innovative approaches and novel datasets. Relevant research findings have been provided in Table 1 that highlight the contributions, applied methods, and drawbacks of those studies. Leukemia classification and progression analysis involve a number of different research technologies, such as machine learning, deep learning, hybrid or ensemble models, and feature fusion. By fusing these methodologies, researchers are often able to gain a comprehensive understanding of leukemia, which facilitates the development of enhanced diagnostic tools and targeted therapies that contribute to improved disease prediction and treatment [22,23]. To efficiently identify many types of leukemia, including healthy, acute lymphocytic leukemia (ALL), acute myeloid leukemia (AML), chronic myeloid leukemia (CML), and chronic lymphocytic leukemia (CLL), Sakthiraj et al. [24] suggested a Hybrid CNN with an Interactive Autodidactic School (HCNN-IAS) technique based on DL technology. The model is trained by using image samples collected from the ASH image back. Even this technique has been applied in an Internet of Medical Things (IoMT) infrastructure, which can be useful in terms of providing services to the patients in their homes. Having a significant recall and precision rate of 99% and an IoMT feature, this approach added a new dimension to the leukemia detection scheme. Another IoMT-based framework entitled “IoMT-Based Automated Detection and Classification of Leukemia Using Deep Learning” was developed by Bibi Nighat et al. [25]. This research enables real-time testing, diagnosis, and treatment of leukemia. The system was built with a combination of cloud computing and medical tools, which are connected to the network sources. However, crucial situations including the COVID-19 pandemic are also supported by this study. DenseNet-121 and ResNet-34 models have been used in this suggested approach to efficiently identify leukemia, where the models work with 99.91% and 99.56% accuracy on the publicly available datasets. For the sake of medical professionals, Dese et al. [26] built a real-time automated diagnostic system using the support vector machine (SVM) to detect four common categories of leukemia. Samples of microscopic blood smear images were obtained from a medical facility in order to train and test the machine learning model (SVM). The method classified all of the leukemia types with an accuracy of 97.69% on test datasets and 97.5% for the validation datasets. Furthermore, in this investigation, the computerized system demonstrated an image processing time of less than one minute, a significantly improved timeframe compared to the manual diagnosis system. Childhood leukemia is one of several stages of leukemia and is also the most challenging to diagnose and treat. Fathi et al. [27] proposed a new technique by the combination of Principal Component Analysis (PCA), neuro-fuzzy (ANFIS), and group method of data handling (GMDH) which can accurately diagnose childhood leukemia. On the basis of a complete blood count (CBC) test, this method basically helps to differentiate between acute lymphoblastic leukemia (ALL) and acute myeloid leukemia (AML). A total of 346 samples were collected, of which 172 contained ALL, 74 contained AML, and 110 were non-patients aged 1–12 years. The approach can be used as an alternative to the presently expensive methodologies. Sridhar et al. [28] developed a modified conventional neural network that can be useful for categorizing malignant and nonmalignant cells. A small dataset was formed using the resources of ASH Bank and ALL-IDB. A data augmentation technique was applied to the dataset to manage those data. The performance evaluation of Resnet-34 and DenseNet-121 encompassed five distinct classes: ALL, AML, CLL, CML, and Healthy. However, the result of the suggested technique was better than the performances of the individual networks, as measured by the testing accuracy of 95.59%. The intricate design of the human body makes it challenging to diagnose B-lymphoblast leukemia or cancer from microscopic samples. To resolve this issue, Kassani et al. [29] devised an automated hybrid method (CNN) based on deep learning that can differentiate between normal tissue and immature leukemic blasts. After augmenting images of the healthy cells and ALL cells, the authors extracted key features from those. Compared with the state-of-the-art models, the developed CNN model performed better in all the performance measurement criteria with an accuracy of 96.17%, 95.17% sensitivity, and specificity of 98.58%. Enhanced deep-learning approaches have a significant contribution to the improvement of leukemia-type classification using precise image processing. One of the popular and critical mutation types in AML is Nucleophosmin 1 (NPM1). Eckardt et al. [30] applied multiple steps of the DL method (CNN) to extract the segmented cells from bone marrow images automatically. Using only extracted images, this approach can classify the mutation status of NPM1, with a significant AUROC value of 0.9699. A total of 1251 patient data were used in this study, and the patients had been newly diagnosed with AML and previously both. Among them, 386 were found with NPM1-muted AML, and 865 patients had NPM1 wild-type AML. Rehman et al. [3] also employed a similar kind of DL approach to categorize ALL into subclasses and figure out the reactive cells from the stained ones. In this research, multiple comparisons were performed between the state-of-the-art models and their developed models. The other classifiers such as Naive Bayesian, KNN, and SVM achieved 78.34%, 80.42%, and 90.91% accuracy, respectively. On the other hand, the proposed approach showed significant performance with 97.78% accuracy for the same dataset. The shape of immature white blood cells (WBCs) in the bone marrow plays an important role in the context of early-stage leukemia identification and treatment. To deal with these issues, Saleem Saba et al. [31] presented an improved deep learning (DL) technique comprising two sequential stages: preprocessing-based classification and segmentation. Deep features are extracted using models like DarkNet-53 and ShuffleNet after synthetic pictures are created using a Generative Adversarial Network (GAN). At the same time, Principal Component Analysis (PCA) is used to extract more useful features and to fuse existing ones. Next, a deep semantic approach combined with morphological processes is used to perform leukemia segmentation. Classification accuracies of 100% and 99.70% were achieved using these methods on the ALL-IDB and LISC datasets, while segmentation accuracies of 99.10% and 98.60% were achieved using the same datasets.

Apart from employing classic machine learning or deep learning approaches, Yadav et al. [32] presented a feature fusion-based deep learning model (SqueezeNet and ResNet-50) that can detect abnormal proliferation in the blood and bone marrow of leukemia-diagnosed patients. The generated model obtained 99.3% classification accuracy using 5-fold cross-validation. For monocyte cells, the ROC value was 100%, and for leukemia-affected cells, it was 99%, demonstrating the model’s excellent sensitivity and resilience.

In another study, Ahmed et al. [33] employed the feature fusion approach in the paper entitled “Hybrid Techniques for the Diagnosis of Acute Lymphoblastic Leukemia Based on Fusion of CNN Features”. Appropriate features were retrieved from the C-NMC 2019 and ALL-IDB2 datasets by combining three CNN models (DenseNet121, ResNet, and MobileNet) and using Principal Component Analysis (PCA). An excellent performance in identifying ALL was noticed by the hybrid model’s DenseNet121-ResNet50-MobileNet-extracted features and RF classifier, which achieved an AUC of 99.1% and an accuracy of 98.8%.

There are clear advantages to using deep learning in the field of medical research, particularly when combined with big datasets. For instance, Aftab et al. [34] introduced an innovative leukemia detection method, employing the Spark BigDL library to analyze microscopic images of human blood cells using a CNN architecture and GoogleNet deep transfer learning. The implemented system exhibited notable improvements, achieving accuracies of 96.42% and 92.69% for training and validation, respectively, in the absence of BigDL. Upon integration with BigDL, however, the training and validation accuracies increased dramatically to 97.33% and 94.78%, respectively, on the same dataset.

N. Ahmed et al. [35] devised a CNN methodology incorporating seven distinct image transformation techniques, demonstrating proficiency in detecting all subtypes of leukemia. The proposed model’s performance was benchmarked against alternative models, including Naive Bayes, SVM, k-nearest Neighbors (KNN), and Decision Trees (DT), utilizing datasets from ALL-IDB and ASH Image Bank. The comparative analysis revealed that the designed CNN model outperformed individual models across performance metrics. Notably, it achieved an accuracy of 88.25% in binary classification (Leukemia vs. Healthy) and 81.74% in multi-class classification (ALL, AML, CLL, CML, and Healthy).

The growing trend of CNNs is noticeable in the classification and diagnosis of medical terms. The training of CNNs, however, necessitates large image datasets. In light of these considerations, Vogodo et al. [36] employed transfer learning for deep feature extraction, utilizing SVM as a classifier, while omitting a segmentation process from the methodology. A new database is introduced in this study by combining three distinct datasets together. The performance of the proposed technique exhibited minor fluctuations across different datasets, achieving 100% classification accuracy on the Hybrid-Leukocyte dataset, 99.20% on the Hybrid-Scale dataset, and 99.76% on the Hybrid-Complete dataset.

## 3. Materials and Methods

### 3.1. The Proposed IoMT Framework

Firstly, all the blood samples are collected from the hospitals. After that, the IoT-enabled microscope uploads the blood smear images to the cloud medical server. In the cloud, the automated diagnosis of leukemia begins with the step of generating the segmented image from the original image with the HSV color thresholding approach. After that, both the original image and the segmented image are sent to the model for classification. The model is developed with the ALL dataset where both the original and segmented images undergo decoding and resizing to a dimension of 128 × 128 in order to ensure uniformity in height and width across all images. This is essential in order to train the model. Then, the values of images ranging from 0 to 255 are normalized to 0 to 1. Subsequently, we employ the data augmentation technique, which involves introducing random shear, random flip, random magnification, and random rotation into the data to enhance the learning capabilities of the model. Following that, we provide the DenseNet-121 model with segmented image data and the VGG16 models with the original image data; these two models then return the features of their respective images, and then both features are combined and transmitted to the classification block in order to classify the class of the image. Finally, the cloud sends the outcome of the model to the medical center and the patient’s personal space. The entire method is shown in Figure 1.

### 3.2. Dataset

The acute lymphoblastic leukemia (ALL) dataset was collected from Kaggle, which is publicly available for research. The dataset was first created in the laboratory of Taleqani Hospital (Tehran, Iran) [37]. After that, this dataset was utilized by Mustafa Ghaderzadeh et al. [38]. In their research work, they generated segmented images for classification. The original images consist of 3256 PBS images obtained from 89 patients suspected of having ALL. The original images are shown in the Figure 2.

In some previous studies, the same number of segmented images is also present in the dataset, and in creating the segmented images, the authors [38] used the HSV color thresholding approach. Some samples of segmented images are shown in Figure 3.

The former consists of hematogenous, while the latter encompasses three subtypes of malignant lymphoblasts: ALL (Early Pre-B), ALL (Pre-B), and ALL (Pro-B).

### 3.3. Image Preprocessing and Segmentation

Image Preprocessing: For data preprocessing, several techniques were used, such as decode and resize, data normalization, and data augmentation. All the images were in JPG format, and the image size was 128 × 128. The pixel value was normalized between 0 and 1. Six transformation techniques were applied to the training data, such as brightness changing, contrast, JPEG noise, and vertical and horizontal rotations, which are randomly applied [38]. These techniques also resolve the most common medical imaging challenges, including class imbalances and noisy data. All the methods and parameters are shown in Table 2.

Image segmentation: A category of embryonic white blood cells known as blast cells serves as an indicator of leukemia. By separating the blast cells from the remaining cells in the blood stain, their counting and analysis are simplified. This may allow for the diagnosis of malignancy and additional blood disorders. The technique implemented in the image is color segmentation-based. This means that the cells in the blood stain can be distinguished from one another by their respective colors. The blast cells in the blood stain are a distinct color from the remaining cells, as seen in the blood cell image. As a result, segmentation via image preprocessing techniques becomes feasible. Before proceeding, the authors performed a color space conversion from the original image to the HSV color space [38]. The HSV color space is more suitable than the RGB color space for the purpose of color segmentation. Two thresholds were subsequently established for the color purple, which is the prevailing hue of blast cells. Finally, a mask was applied to the image in order to isolate the blast cells from the remainder of the cells. We implemented the entire process in our cloud, which generated the segmented image automatically and transmitted it to the leukemia classification model.

### 3.4. Adopted Transfer Learning Models

DenseNet-121 Model

The CNN architecture in question was initially proposed by Huang et al. [39] in their publication. The DenseNet-121 model comprises four dense blocks, each of which is connected in a manner that permits it to receive information from all preceding layers. The model consists of 121 layers. As a result, the model is exceptionally effective at feature extraction from images. The effectiveness of the DenseNet-121 model has been demonstrated across a range of image classification applications, such as facial recognition, object detection, and image segmentation. Transfer learning is another prevalent application in which a model trained on a substantial dataset of images is utilized to initialize the weights of a new model being trained on a more limited dataset [40]. DenseNet-121 comprises four dense nodes in total. Every dense block is composed of several convolutional layers that are interconnected to enable each layer to obtain information from all preceding layers. As a result, the model is exceptionally effective at feature extraction from images.

VGG16 Model

We utilized a VGG16 architecture to construct our CNN model in our research. The designation “VGG16” denotes the precise arrangement of sixteen layers [41,42]. The design of this architecture is distinguished by its uncomplicated implementation of layered convolutional layers. In order to enhance model performance and accelerate convergence, pre-existing weights from the ImageNet dataset were integrated into the training procedure. ImageNet, a significant dataset comprising a wide variety of annotated images, furnished the VGG16 model with fundamental characteristics and depictions [43]. For computer vision tasks, the deliberate use of a pre-trained VGG16 model with ImageNet weights is a critical component of our transfer learning strategy. The pre-existing weights encompass general characteristics that are pertinent to a wide range of visual recognition tasks.

### 3.5. Architecture of Deep Feature Fusion

For feature extraction from the input images, we employed the DenseNet-121 and VGG16 transfer learning models, both of which are capable of feature extraction autonomously. Both of the models are pre-trained on extensive datasets, such as ImageNet. Their combined capabilities can capitalize on the advantages of pre-trained weights, thereby improving transfer learning effectiveness. VGG16 models capture the high details and hierarchical features of the images, while the DenseNet-121 model captures the intricate patterns and relationships as it has dense connectivity [39,41]. For this reason, the combination of these two models is performed to obtain the best features from both original and segmented images. Figure 4 shows the overall model architecture.

In order to obtain an equivalent number of features, we extract the conv4_block9_0_bn layer from the VGG16 model and the block5_conv3 layer from the DenseNet-121 model. It is then necessary to concatenate every feature. The classification is performed utilizing dense and dropout layers. The model receives 128 × 128 features as input, and the DenseNet-121 and VGG16 models both return 512 features in an 8 × 8 format. By connecting each feature map directly to the output layer, we are able to preserve spatial information and reduce parameters while calculating the average value of each feature map across its entire spatial dimension using Global Average Pooling2D. When the 512 features of both models are combined and passed to the dense layers, this method improves translation invariance, functions as regularization, obviates the necessity for a flattening layer, and achieves computational efficiency.

An initial dense layer consisting of 1024 units is defined. After the dense layer, a dropout layer is implemented, characterized by a dropout rate of 0.2. In neural networks, the dropout layer is a prevalent regularization technique used to prevent overfitting. A dropout rate of 0.2 indicates that each neuron in the previous layer has a 20% chance of being “dropped out” or reset to zero during each update during training. This mechanism aids in mitigating the model’s excessive dependence on particular neurons and promotes more resilient learning through the prevention of hidden unit co-adaptation. The dense layer unit is then decreased by half, and a dropout layer with a rate of 0.2 is implemented once more. Subsequently, six additional dense layers are implemented, with each reducing in unit value by half. Using dense layers with a halving method allows for the model to learn complex patterns while retaining computing efficiency and avoiding overfitting. The final dense layer contained four units, which are equal to our class numbers.

The connection and flow diagram of the model’s overall strata is illustrated in Figure 5. The figure illustrates the complete connectivity of each layer, along with the names of the layers. The diagram also provides a clear understanding of the model’s architecture. The proposed model summary is shown in Table 3. It shows the layers of a neural network model, their output shapes, the number of parameters in each layer, and the layers each layer is connected to. The first two rows of the table show the input layers to the model. These are called input_original and input_segmented. They both have an output shape of (None, 128, 128, 3), and each batch element is a 128 × 128 × 3 tensor (the dimensions refer to height, width, and channels, respectively). The next two rows show the two main branches of the model. The model branch takes input_original as input, and the model_1 branch takes input_segmented as input.

Both branches then go through a series of convolutional and pooling layers and eventually output a 512-dimensional vector. The two branches are then concatenated together, and the resulting 1024-dimensional vector is fed into a series of dense layers. The final layer in the model is a dense layer with four outputs, which means that the proposed model predicts a 4-dimensional vector as its output. The table also shows the number of parameters in each layer. The total number of parameters in the model is 18,598,836, of which 1,749,556 are trainable parameters. This means that the model has a lot of capacity to learn complex relationships in the data.

### 3.6. Hyperparameter Tuning

Understanding the importance of fine-tuning the hyperparameters of a deep learning (DL) model is essential as it directly impacts the model’s learning process and overall performance. In each experimental trial, a batch size of 32 is used, and the epoch is configured to last for 50 iterations. The model includes normalization and augmentation strategies to address the potential issue of overfitting. The use of two dropout layers, each with a dropout rate of 0.2, is implemented inside the classification block in order to augment the efficacy of our model. A learning rate of 0.001 is used. The optimizer used in this study is Adam, whereas the loss function utilized is sparse categorical cross-entropy.

## 4. Results

### 4.1. Experimental Setup

The weights of the pre-trained DenseNet-121 and VGG16 models were initialized with the pre-trained ImageNet weights and remained frozen during the feature extraction process. To accomplish the final categorization, the training was limited to the newly added completely connected layers because only those could be updated. This approach uses the powerful feature extraction capabilities of the pre-trained models while allowing for the new layers to learn strong patterns.

The dataset was partitioned into a ratio of 7:2:1, with 70% of the data allocated for training the model. A validation set of 10% of the data was used, while the remaining data were allocated for testing purposes. A total of 2279 images were used for training, 652 images were used for testing, and 325 images were used for the validation of the model. The experiment that was conducted as part of our study is detailed below.

The VGG16 model was trained using only original images, segmented images, and a combination of the two.The DenseNet-121 model was trained using only original images, segmented images, and a combination of the two.The proposed model was trained using only original images, segmented images, and a combination of the two.

Following these experiments, we conducted a comparative analysis of the outcomes and classification matrices of all the models.

### 4.2. Environment Setup

The TensorFlow Keras package was used for classification, and the model ran on a Kaggle environment with GPU T4 X2 for acceleration and 13 GB RAM. All computational tasks in the proposed system were performed in Python. The fusion model was evaluated using accuracy, precision, recall, F1-measure, specificity, and a confusion matrix. True Negative *(TN)*, *True Positive (TP)*, *False Negative (FN)*, and *False Positive (FP)* represent the model’s performance in the confusion matrix. The performance metrics utilized in this study are listed below:

Accuracy:(1)$$Accuracy=\frac{TP+TN}{TP+TN+FP+FN}$$

Precision:(2)$$Precision=\frac{TP}{TP+FP}$$

Recall:(3)$$Recall=\frac{TP}{FN+TP}$$

Specificity:(4)$$Specificity=\frac{TN}{TN+FP}$$

F1 measure:(5)$$F_{1}=2\times \frac{Precision\times Recall}{Precision+Recall}$$

Accuracy, precision, recall, specificity, and F1-score are essential metrics for evaluating medical imaging analysis because they provide a detailed and sophisticated understanding of the model’s performance, ensuring both its reliability and effectiveness in real-world medical settings. The accuracy of the fusion model reflects its overall correctness, while precision evaluates the accuracy of positive predictions and specificity gauges the ability to correctly identify negative instances. However, recall assesses the capacity to identify all positive instances, while the F1-score finds a balance between precision and recall. These metrics offer a holistic perspective and aid in our understanding of the performance of the feature fusion model to classify leukemia.

### 4.3. Learning Curve of Original Images

The first graph in Figure 6, titled “Model Accuracy”, depicts the model’s performance in terms of accuracy during the training process. The graph reveals a positive trend, with both training and validation accuracy curves reaching close to 100%. This indicates that the model effectively learns from the training data and generalizes well to unseen data.

The second graph seen in Figure 6, labeled as “Model Loss”, shows the progression of the model’s loss during 50 epochs. This graph shows two separate lines, with the orange line representing the model loss and the blue line representing the training validation. Both lines have a consistent downward trajectory, suggesting a reduction in loss with an increasing number of epochs. The model’s loss constantly stays lower than the train validation, indicating that the model successfully learns from the training data and improves its capacity to reduce mistakes.

### 4.4. Learning Curve of Segmented Images

The first graph in Figure 7, labeled as “Training and Validation Accuracy Curve”, presents a comparison of the model’s accuracy in both training data (train) and validation data (valid) during a span of 50 epochs. Both lines have a positive trajectory, suggesting that the model acquires knowledge and improves its precision. Nevertheless, it is seen that the accuracy of the training data continuously surpasses that of the validation data, indicating a potential case of overfitting to the training dataset.

The second graph in Figure 7, labeled as “Training and Validation Loss Curve”, illustrates the loss of the model throughout 50 epochs. It is visually displayed by two lines, with the orange line indicating the model loss and the blue line representing the training validation loss. Both lines demonstrate a consistent decrease, suggesting that the model effectively minimizes its mistakes as it acquires knowledge. The model’s loss constantly stays lower than the training validation, indicating that the model successfully acquires knowledge from the training data.

### 4.5. Learning Curve of Combined Images

Figure 8’s Training and Validation Accuracy Curve illustrates the performance of the model on both training and validation data, indicating a favorable upward trend in accuracy. As the number of epochs is increased, the model demonstrates an enhanced capacity to acquire knowledge and provide precise predictions, as shown by the upward trajectory of the train accuracy curve. The validation accuracy curve has a favorable trajectory, suggesting that the model’s ability to perform on unknown data progressively improves. This observation indicates that the model exhibits good generalization and does not suffer from overfitting to the training data. In general, the graph indicates that the model exhibits effective learning capabilities and attains sufficient levels of accuracy on both the training and validation data.

In the loss curve, both the training and validation curves have a consistent downward trajectory, suggesting a reduction in loss as the number of epochs progresses. The model regularly exhibits a lower loss compared to the training validation, indicating that it efficiently learns from the training data. The observed pattern indicates that the model progressively improves its capacity to mitigate mistakes and provide precise forecasts.

### 4.6. Confusion Matrix

Figure 9 represents the confusion matrix of the original, segmented, and combined images. The model demonstrates a high level of performance when both the original and segmented images are supplied. Out of the original photos, a total of five misclassifications are observed. Among them, four images are erroneously labeled as benign instead of early, while one image is mistakenly classified as benign instead of pro class. The model misclassifies a total of seven photos when just segmented images are supplied. The majority of these misclassifications occur inside the benign class. One picture is categorized as pre, while four photographs are misclassified as pro instead of belonging to the benign class.

The use of both photos in the analysis results in a more precise classification of the model, as seen by the combined confusion matrix. A total of two photos are erroneously identified as belonging to the benign class and the pre-class.

### 4.7. Performance Comparison

Our model is evaluated in comparison to the DenseNet-121 and VGG16 models using both categories of images. DenseNet-121 and VGG16 are also assessed by utilizing our approach, which involves the combination of images.

A variety of transfer learning models were also employed in an experiment to test various combinations. Nevertheless, Table 4 demonstrates that the combination of the VGG16 and DenseNet-121 models obtains the highest level of accuracy.

Combining multiple features into a single representation can greatly enhance performance in image detection and data analysis tasks. By incorporating multiple features, models can better capture complex patterns and relationships that may not be immediately evident while using solo features alone. Moreover, through the integration of various features, the model is able to effectively filter out irrelevant noise and prioritize the most pertinent information. A 96.62% accuracy is obtained by the VGG16 model using only original images. By exclusively utilizing segmented images in the VGG16 model, the accuracy is marginally reduced to 95.38%. However, by combining both types of images and extracting features using VGG16, the accuracy is significantly enhanced to 98.95% when classifying leukemia. A similar method is employed for DenseNet121, yielding an enhanced accuracy of 99.08% when applied to combined images. However, the DenseNet121 model illustrates better performance compared to VGG16 in all three cases. The proposed fusion model, which aggregates features from two solo models DenseNet-121 and VGG16, outperforms in all comparisons with original, segmented, and combined image types, achieving accuracies of 98.46%, 97.85%, and 99.89%, respectively.

All measurements suggest that training the model with both original and segmented images performs better than using only segmented or original images, as shown in Figure 10. Precision, recall, and F1-score metrics are employed to assess each model’s effectiveness, with a specific emphasis on the impact of training with combined images, which are shown in Table 5. The results consistently reveal that when models are trained with combined images, there is a notable enhancement in precision, recall, and F1-score across all evaluated models. For instance, in the case of VGG16, the precision significantly improves from 96.69% (original) and 95.15% (segmented) to 97.86% when trained with the combined images. Similar trends are observed for DenseNet-121 and the proposed model, showcasing the generalizability of this improvement across different architectures.

Notably, the proposed model consistently outperforms both VGG16 and DenseNet-121 in every scenario, with precision, recall, and F1-score peaking at 99.803%, 99.89%, and 99.76%, respectively, when trained with combined images. These findings underscore the significant impact of employing combined images in model training, with the proposed model emerging as the preferred choice for image classification tasks, particularly when considering the combined dataset. This study contributes valuable insights into the optimization of model training strategies for enhanced classification performance across diverse data representations.

## 5. Discussion

We demonstrate that the accuracy is not particularly high when training the VGG16 model or the DenseNet-121 model in isolation. However, by combining these two models, we are able to attain a higher score. It demonstrates that the results obtained from combining these two models for feature extraction are enhanced. To show the significant outcomes, the proposed model of this study is compared with two of the other existing studies. The first comparison is with the study of Mohamed E. Karar et al. [44], where the ASH image bank is used. In this study, the proposed model achieves an accuracy of 99.89%. The proposed model demonstrates superior accuracy compared to the study conducted by Mohamed E. Karar et al. [44], wherein the authors achieved an accuracy of 99.58%. Beyond accuracy, the proposed model outperforms the study in all other metrics. Figure 11 illustrates that the proposed model in this research attains a precision of 99.80%, a recall of 99.72%, a specificity of 99.89%, and an f1-score of 99.76%. In contrast, the study by Mohamed E. Karar et al. [44] reports comparatively lower results: 96.67%, 94%, 98.74%, and 95% for precision, recall, specificity, and F1-Score, respectively.

Compared with another existing work, the accuracy of the proposed model is marginally higher at 99.89%, which is only a fraction higher than the 99.85% achieved by Mustafa Ghaderzadeh et al. [38], which is shown in Figure 11. When considering precision, the proposed model demonstrates a slightly better performance of 99.8% in comparison with the 99.74% achieved by Mustafa Ghaderzadeh et al. The marginal improvement in accuracy suggests that the proposed model may identify positive instances more precisely. Likewise, regarding recall, the proposed model surpasses Mustafa Ghaderzadeh et al.’s once more, attaining a rate of 99.72% as opposed to 99.52%. This indicates that the proposed model has an increased capacity to identify positive cases.

The careful equilibrium achieved by the proposed model is apparent in its F1-Score of 99.76%, which is slightly higher than the 99.63% achieved by Mustafa Ghaderzadeh et al. The proposed model demonstrates a nuanced superiority, indicating that it not only achieves high precision but also handles false positives and false negatives with efficacy. Although numeric differentials are nuanced, they possess considerable importance within the domain of classification models. The proposed model, through its gradual improvements, presents itself as a highly prospective candidate for implementation in practical settings. Without any subjective judgments, the numerical data present an account of advancements and possibilities, encouraging additional contemplation regarding the pragmatic ramifications of these subtle enhancements.

A comparative analysis of thirteen recent studies on the classification of blood cell images is provided in Table 6, which includes a summary of the dataset sources, classifiers utilized, and accuracy percentages. Significantly, the proposed classification model for blood cell images, which utilizes DenseNet-121 and VGG16, attains the maximum reported accuracy of 99.89% when applied to the ALL image dataset. Additional research conducted by Saleem Saba et al. and Sakthiraj et al., likewise, exhibit noteworthy levels of accuracy, precisely 99.70% and 99%, respectively. The table emphasizes the variety of approaches utilized and the efficacy of sophisticated models in improving the precision of image classification systems for blood cells.

Overall, the performance outcomes underscore the efficacy of the model proposed in this study. A comparative analysis against related existing methodologies within the domain of leukemia detection, particularly those employing feature fusion techniques, reveals the superior performance of the proposed model. The comprehensive evaluation conducted on original, segmented, and combined image types is instrumental in addressing rising research questions in this field. Moreover, a demo implementation is shown with the concept of using AWS cloud service in an MIoT-based framework. The applicability of the proposed technique extends beyond leukemia or blood cancer diagnosis, demonstrating potential utility in the detection of various diseases diagnosed through the analysis of Complete Blood Count (CBC) or blood cells. The proposed framework can be useful in different ways in medical sectors as described below:Automated Leukemia Detection: Automated leukemia detection offers a range of benefits for patients, including early detection and intervention, increased accuracy and consistency in diagnosis, reduced turnaround time for results, and the potential for personalized treatment approaches. These advancements contribute to improved patient outcomes, an enhanced quality of life, and a more streamlined and effective healthcare process for individuals facing leukemia.Medical IoT Devices: The suggested method, applied to medical IoT devices, can significantly enhance their functionality and utility in healthcare. Integrating deep learning methods into medical IoT devices enhances their capacity for data analysis, pattern recognition, and predictive analytics. This, in turn, leads to improved diagnostic accuracy and more proactive healthcare management for patients. The continuous learning capabilities of deep learning models contribute to the ongoing refinement and optimization of healthcare interventions based on real-world data.Health Science Research: The concept of an automated leukemia detection system can be extended to health science research and various challenging diseases. By leveraging advanced technologies, such as artificial intelligence and deep learning, this concept can be applied beyond leukemia to a wide range of challenging diseases. The key applications include the development of multi-disease detection platforms, integration of multi-omics data for a holistic understanding of diseases, public health surveillance, drug discovery and development, continuous monitoring with real-time feedback, and the incorporation of deep learning in healthcare research.

## 6. TestBed Implementation of The Proposed Method

We implemented the system on the AWS cloud platform due to its superior dependability and economical pricing. We utilized the most widely used S3 bucket for storage and Lambda to execute the model and process within the AWS cloud. Blood sample images uploaded by users are stored directly in the S3 container, and the user is notified via affirmation that the image has been uploaded. The lambda function then invokes the S3 container containing the binary image of the testing image after receiving the blood image key of that image. Lamba then initiates the procedure during which the proposed model is executed and a segmented image is produced from the test image. By utilizing these two distinct categories of images, the model generates a forecast, which is subsequently transmitted to the user’s personal devices. The flow diagram of the proposed method is shown in Figure 12. Figure 13 illustrates a rudimentary web application through which users are able to submit blood images in order to receive predictions and recommendations. The initial image output indicates that the subject is benign, with a 90 % probability, whereas the probabilities for the other classifications are negligible; consequently, the system returns a positive message. Conversely, with a maximum probability of 95 %, the second image is categorized as pre, a subtype class of leukemia; consequently, the system advises users to consult an expert doctor.

## 7. Conclusions and Future Works

The proposed AI-based IoMT framework can classify leukemia automatically with the combination of IoMT devices and a cloud server where a fusion model classifies leukemia by generating segmented images from the given original images. By employing transfer learning models like VGG16 and DenseNet-121, we were able to construct a model that exhibits superior accuracy compared to the majority of alternative approaches. The model is provided with two image inputs: the original image and segmented versions of the original. Both models independently extract features from the images; however, we must ensure that both models produce an equivalent number of features when applied to two distinct categories of images. Certain methods require authors to work exclusively with either original or segmented images, which may result in the omission of valuable features from the images. In addition, we demonstrate that the performance of the VGG16 and DenseNet-121 models combined to classify leukemia using original, segmented, or combined images is superior to that of either model used alone. Our research demonstrates that the performance of the model is enhanced when both the original and segmented images are utilized. The model’s accuracy is 99.89% when both the original and segmented images are provided; however, its accuracy decreases when only one form of image is utilized. Given its high accuracy in classifying leukemia from MRI images, it is highly probable that this technology could be implemented in the medical field to detect the disease in its earliest stages in patients. In the future, we intend to tune the model further in order to increase its accuracy. The current dataset consists of 6512 images combined, both original and segmented, which is insufficient for the robustness of the model, so we intend to train the model with more images. Additionally, the model is not tuned; therefore, in the future, hyper-tuning may improve the model’s accuracy.

## Figures and Tables

**Figure 1 sensors-24-04420-f001:**
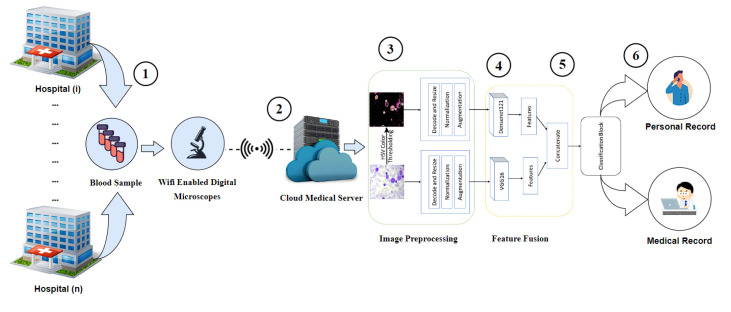
Workflow of the proposed framework. This workflow comprises six essential components: 1. Image acquisition 2. Cloud-based feature fusion model. 3. Image preprocessing, 4. Extraction of features 5. Block for concatenating features and classifying them. 6. Sending the outcome to the medical center and patient.

**Figure 2 sensors-24-04420-f002:**
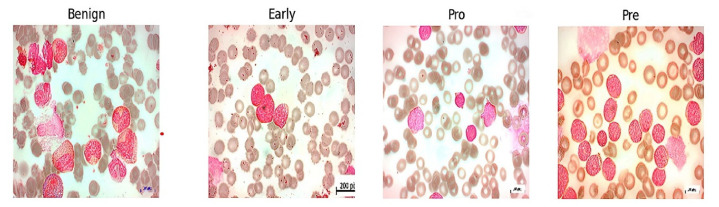
Original images.

**Figure 3 sensors-24-04420-f003:**
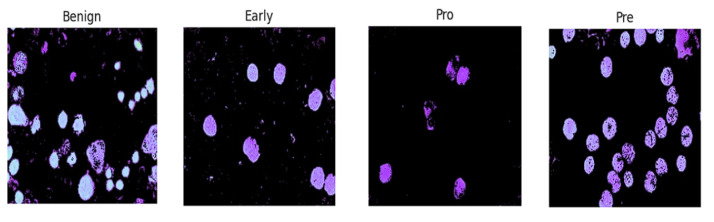
Segmented images.

**Figure 4 sensors-24-04420-f004:**
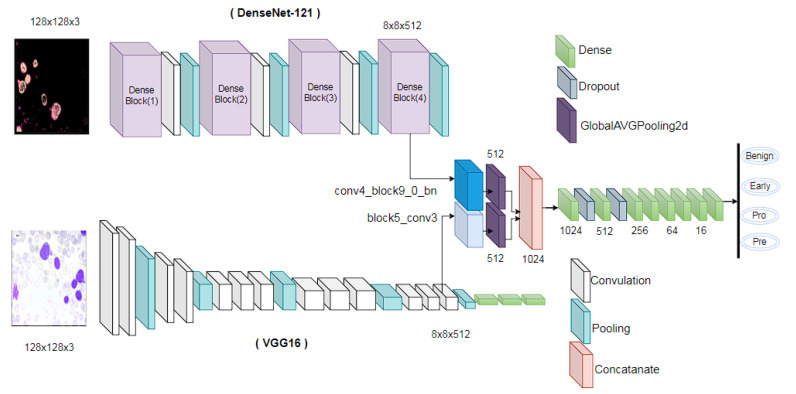
The structure of the suggested model. The input image shapes are 128 × 128 × 3, and feature extraction is performed using transfer learning models. In order to reduce the number of parameters and preserve spatial information, the global average pooling2D is utilized. To mitigate overfitting concerns, dropout layers with a value of 0.2 are implemented in the dense layer.

**Figure 5 sensors-24-04420-f005:**
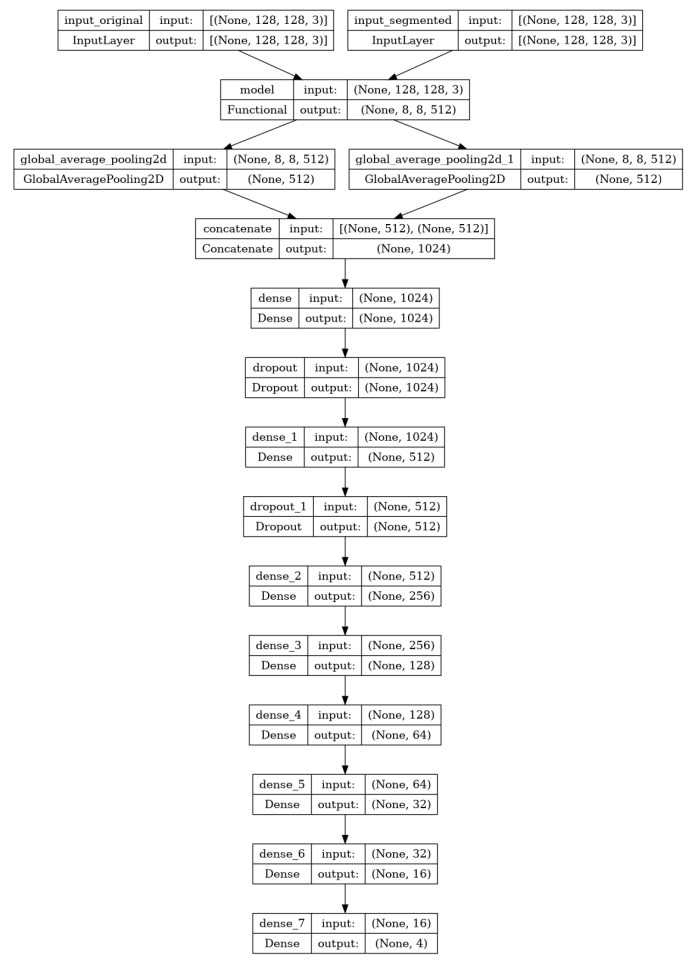
Leveraging spatial and morphological features: A squeeze-and-excitation enhanced deep learning architecture for leukemia classification.

**Figure 6 sensors-24-04420-f006:**
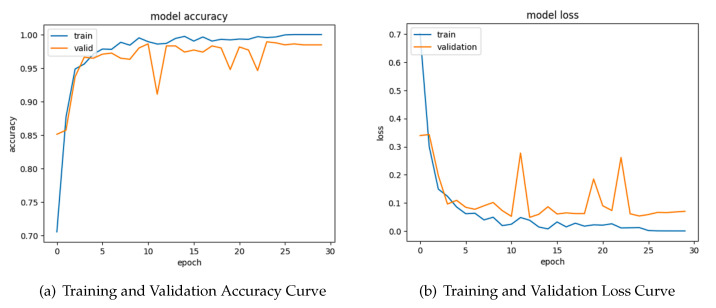
Original images’ training and validation.

**Figure 7 sensors-24-04420-f007:**
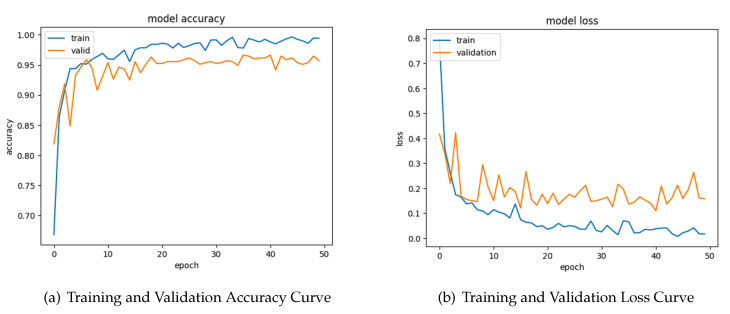
Segmented images’ training and validation.

**Figure 8 sensors-24-04420-f008:**
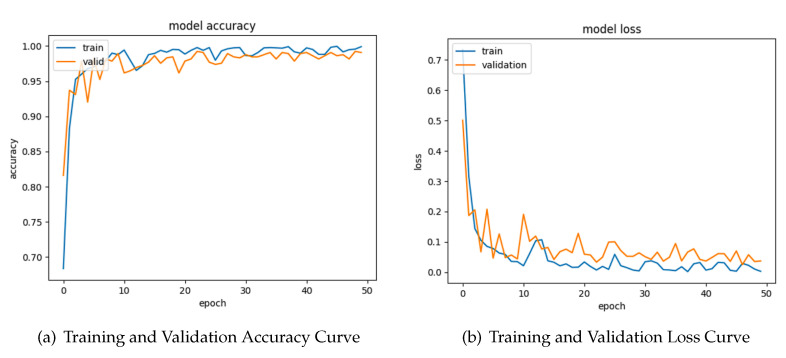
Combined images’ training and validation.

**Figure 9 sensors-24-04420-f009:**
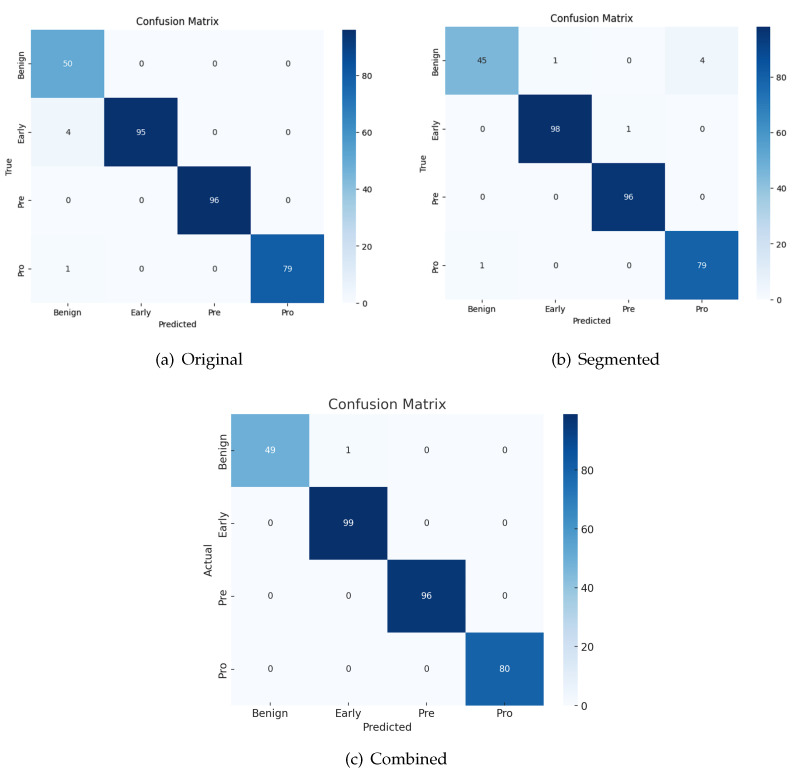
Confusion matrix.

**Figure 10 sensors-24-04420-f010:**
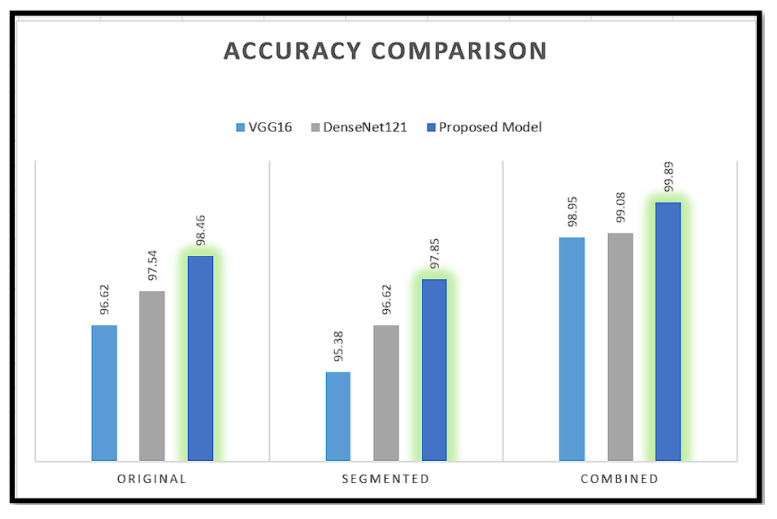
Comparison between traditional CNN models.

**Figure 11 sensors-24-04420-f011:**
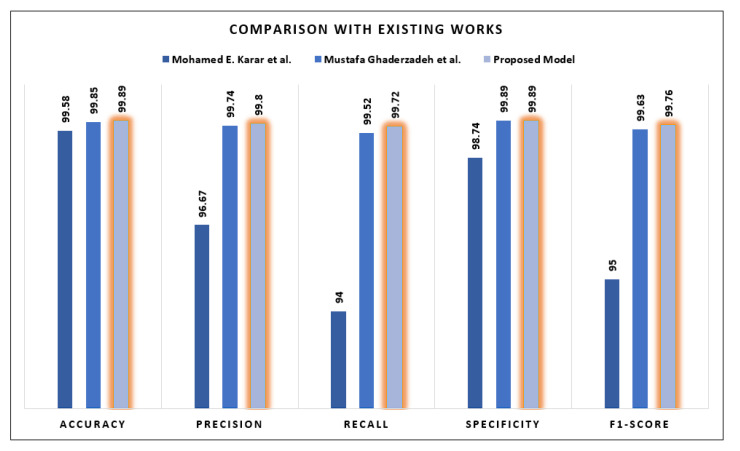
Comparison between Mohamed E. Karar et al. [44] and Mustafa Ghaderzadeh et al. [38].

**Figure 12 sensors-24-04420-f012:**
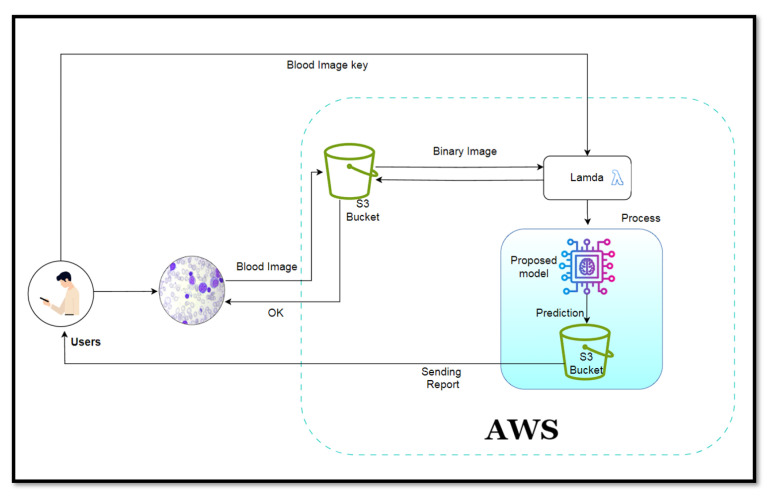
Flow Diagram of classification process in the AWS Cloud Server. A patient or user can upload their sample test image in the cloud server; the server trained with the deep learning model can perform image preprocessing and testing. Finally, it sends the notification of the results to the user.

**Figure 13 sensors-24-04420-f013:**
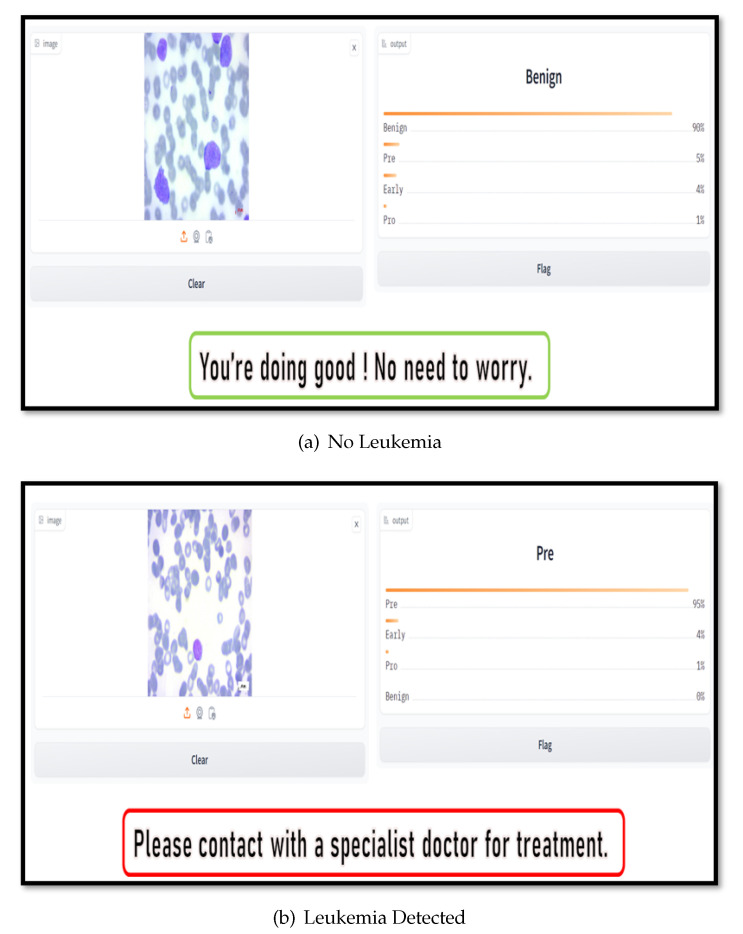
Leukemia Classification Web Application. A user uploads the sample images for prediction, all the processing is then performed in the cloud server, and the results of the sample images are then sent back to the user.

**Table 1 sensors-24-04420-t001:** Existing works with key technologies and drawbacks.

Authors	Contribution	Applied Method	Drawbacks
Rehman et al. [3]	Proposed a DL method based on transfer learning	CNN	Accuracy is inadequate
Sakthiraj et al. [24]	Implemented a hybrid model on an IoMT platform that performs feature extraction, fusing, and classification	HCNN-IAS	Only authentic images were utilized
Bibi Nighat et al. [25]	Included cloud computing in an IoMT-based leukemia detection framework	ResNet-34 and DenseNet-121	Require the ability to diagnose subcategories of each form of leukemia.
Dese et al. [26]	Real-time automated leukemia diagnostic system (process time less than 1 min)	WBC segmentation, Feature extraction	The dataset was not enough
Sridhar et al. [28]	Choosing effective key features using ML techniques	Resnet-34 and DenseNet-121	Accuracy is not satisfactory
Kassani et al. [29]	Applied various data augmentation techniques to avoid over-fitting problems	VGG16 and Mobilenet	Segmented images alone were employed
Yadav et al. [32]	Proposed a feature fusion-based DL method	SqueezeNet and ResNet-50	The absence of segmented images results in an inadequate level of accuracy
Ahmed et al. [33]	Developed a hybrid model using three CNN models for extracting features and used ML models for classification	DenseNet121-ResNet50-MobileNet for feature extraction and RF-XGB for classification	Insufficient image data caused overfitting.

**Table 2 sensors-24-04420-t002:** Augmentation methods and parameters.

Methods	Parameters
Brightness	[−5%,+5%]
ine Contrast	[−8%,+8%]
ine Rotation	$\left[-15^{\circ },+15^{\circ }\right]$
JPEG noise	[30,100]
Flip	Horizontal, Vertical

**Table 3 sensors-24-04420-t003:** Model summary.

Layer (Type)	Output Shape	Param #	Connected to
input_original (InputLayer)	[(None, 128, 128, 3)]	0	[ ]
input_segmented (InputLayer)	[(None, 128, 128, 3)]	0	[ ]
model (Functional)	(None, 8, 8, 512)	14714688	[’input_ original[0][0]’]
model_1 (Functional)	(None, 8, 8, 512)	2134592	[’input_segmented[0][0]’]
global_average_pooling2d (GlobalAveragePooling2D)	(None, 512)	0	[’model[0][0]’]
global_average_pooling2d_1 (GlobalAveragePooling2D)	(None, 512)	0	[’model_1[0][0]’]
concatenate (Concatenate)	(None, 1024)	0	[’global_average_pooling2d[0][0]’, ’global_average_pooling2d_1[0][0]’]
dense (Dense)	(None, 1024)	1049600	[’concatenate[0][0]’]
dropout (Dropout)	(None, 1024)	0	[’dense[0][0]’]
dense_1 (Dense)	(None, 512)	524800	[’dropout[0][0]’]
dropout_1 (Dropout)	(None, 512)	0	[’dense_1[0][0]’]
dense_2 (Dense)	(None, 256)	131328	[’dropout_1[0][0]’]
dense_3 (Dense)	(None, 128)	32896	[’dense_2[0][0]’]
dense_4 (Dense)	(None, 64)	8256	[’dense_3[0][0]’]
dense_5 (Dense)	(None, 32)	2080	[’dense_4[0][0]’]
dense_6 (Dense)	(None, 16)	528	[’dense_5[0][0]’]
dense_7 (Dense)	(None, 4)	68	[’dense_6[0][0]’]
Total params:	18,598,836		
Trainable params:	1,749,556		
Non-trainable params:	16,849,280		

**Table 4 sensors-24-04420-t004:** Performance metrics for different feature fusions.

Experimental Models	Accuracy (%)	Sensitivity (%)	Precision (%)	Specificity (%)
DenseNet121-ResNet50	98.9	99.2	97.8	94.7
DenseNet121-MobileNet	99.2	98.8	98.6	97.4
Proposed Model	99.87	98.80	99.803	99.72

**Table 5 sensors-24-04420-t005:** Classification report of all models.

Models	Data Type	Precision	Recall	F1-Score
VGG16	Original	96.69	96.23	96.45
VGG16	Segmented	95.15	94.95	94.97
VGG16	Combined	97.86	96.99	97.38
DenseNet-121	Original	97.54	97.53	97.5
DenseNet-121	Segmented	96.62	96.61	96.43
DenseNet-121	Combined	99.1	99.09	99.09
Proposed Model	Original	98.6	98.4	98.48
Proposed Model	Segmented	97.86	97.84	97.82
Proposed Model	Combined	99.803	99.72	99.76

**Table 6 sensors-24-04420-t006:** Comparison of different methods.

Authors	Dataset	Classifier	Accuracy (%)
Rehman et al. [3]	Collected from Amreek Clinical Lab-oratory Saidu Sharif Swat KP Pakistan	CNN	97.78
Dese et al. [26]	Collected from Department of Hematology, Jimma University Medical Center (JMC)	MCSVM	97.69
Sridhar et al. [28]	ASH image bank and ALL-IDB	Resnet-34 and DenseNet-121	95.59
Kassani et al. [29]	ISBI[2019]	CNN	96.17
Saleem Saba et al. [31]	ALL-IDB and LISC	SVM-KNN	99.70
Sakthiraj et al. [24]	ASH image bank	HCNN-IAS	99
Bibi Nighat et al. [25]	ALL-IDB and ASH image bank	ResNet-34 and DenseNet-121	99.56 and 99.91
Yadav et al. [32]	Blood Cell Images	SqueezeNet and ResNet-50	99.3
Ahmed et al. [33]	C-NMC[2019] and ALL-IDB2	DenseNet121-ResNet50-MobileNet	98.8
Aftab et al. [34]	Collected from The American Society of Hematology	Apache Spark BigDL using GoogleNet	94.78
N Ahmed et al. [35]	ALL-IDB and ASH Image Bank	CNN	88.25 and 81.74
Vogodo et al. [36]	ALL-IDB1,ALL-IDB2,Leukocytes, CellaVision	AlexNet, CaffeNet, and Vgg-f	99.20
Proposed Model	ALL image dataset [38]	DenseNet-121 and VGG16	99.89

## Data Availability

The selected dataset was sourced from a free and open access source Kaggle, leukemia Dataset: https://www.kaggle.com/datasets/mehradaria/leukemia (accessed on 5 January 2024).

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
