# Peer review of "Utilizing Deep Feature Fusion for Automatic Leukemia Classification: An Internet of Medical Things-Enabled Deep Learning Framework"

_sensors, 2024, doi:10.3390/s24134420_

Round 1

Reviewer 1 Report

Comments and Suggestions for Authors

The paper presents neural network architecture for automatic Leukemia classification. The architecture combines the features obtained from DenseNet-121 (fed with segmented image) and VGG16 (fed with original image). It is demonstrated that the proposed method performs better than the compared approaches.

My suggestions and comments are as follows:

1. This work indicates that the combination of DenseNet-121 and VGG16 can improve the classification accuracy. It would be interest to check if the combination of other models also validates this finding.

2. The first and last contributions can be combined and shortened to one, serving as the third contribution.

2. The sizes of the words and numbers in Fig. 9 are too small for reading.

Author Response

Please see the attached file where We have responded to all the comments raised by the reviewers.

Reviewer 2 Report

Comments and Suggestions for Authors

How does the proposed model handle potential challenges such as class imbalances, noisy data, or image quality variations that are common in medical imaging datasets? Can you explain the dataset's preprocessing techniques?

How were the hyperparameters of the deep learning models tuned to optimize performance, and what considerations were taken into account during this process? Can you add the details in the appropriate section?

How is the proposed fusion model's performance compared to individual models that were trained solely on original or segmented images, and what insights were gained from this comparative analysis? Please add these insights in the comparison section.

Need to add more details on the experimental setup in terms of data partitioning for training, validation, and testing, as well as any cross-validation techniques employed during model evaluation.

Please address the following questions,

What criteria were used to evaluate the proposed fusion model's performance, and how did these metrics align with the research study's objectives?

How were these architectures (DenseNet-121 and VGG16 models) chosen for feature extraction in the proposed framework?

How were the weights of the pre-trained DenseNet-121 and VGG16 models initialized and fine-tuned during the feature extraction process?

How were the feature maps extracted from the DenseNet-121 and VGG16 models combined before feeding them into the classification block, and what advantages did this feature fusion approach offer in terms of model interpretability and performance?

Comments on the Quality of English Language

Grammatical errors are there for eg, in the following line numbers 154, 300, Figure 13 description,

Author Response

Dear Reviewer,

Please find the review reply in the following file.

Reviewer 3 Report

Comments and Suggestions for Authors

The study introduces an innovative AI-based Internet of Medical Things (IoMT) framework aimed at improving the diagnosis of Acute Lymphoblastic Leukemia (ALL), a cancer affecting blood and bone marrow. Traditional diagnosis methods like blood tests and biopsies are time-consuming and costly, making early diagnosis challenging. The proposed IoMT system automates leukemia detection using peripheral blood smear (PBS) images, transmitting diagnostic reports to a centralized database. Blood samples are collected, and PBS images are sent to a cloud server via a WiFi-enabled microscope. The system utilizes a fusion model trained on 6512 images from 89 individuals, employing VGG16 for original image feature extraction and DenseNet-121 for segmented images. The combined features are then used for leukemia classification, achieving remarkable accuracy, precision, and recall rates, significantly outperforming existing CNN models. Additionally, a web application was developed for simulation purposes, showing great promise for computer-aided leukemia detection in biomedical research.

However, I have the following queries. 

Abstract:

artificial intelligence may be artificial intelligence (AI)

also, give the full form of "ALL."

1. Line number 37:

Please provide the citation for the American Cancer Society.

2. Line numbers 77 to 79:

A pre-trained customized AlexNet was developed to classify ALL while overfitting was reduced via data augmentation and This (please replace it with "this") research attained a level of accuracy up to 96.06%[11].

3. Line numbers 93 to 95:

The DenseNet-121 model was employed to extract the features from the segmented images, while VGG16 has been customized with the original ALL images. 

Please provide the citations of densenet-121 and VGG 16.

4: Literature Review.

Could you please provide the full forms of "ALL," "AML," and "CML"? Line numbers 159 to 161 

To efficiently identify many types of leukemia, including healthy, ALL, AML, CML,  and CLL, Sakthiraj et al. [18] suggested a Hybrid Convolutional Neural Network with an Interactive Autodidactic School (HCNN-IAS) technique based on DL technology.

5. Line number 175:

support vector machine. It could be a support vector machine (SVM). 

6. Please improve the figure 1. Improvement in terms of visualization. Please increase the font size, bold it, etc. 

7. Line numbers 328 to 330. Please add a proper citation for this sentence. 

Transfer learning is another prevalent application in which a model trained on a substantial dataset of images is utilized to initialize the weights of a new model being trained on a more limited dataset.  

8. Line numbers 368 to 370. Could you please explain: what are the advantages of utilizing six additional layers

Subsequently, six additional dense layers were implemented, with each reduction in unit value by half; the final dense layer contained four units, which is equal to our class numbers.

Author Response

Dear Reviewer,
